# Enhancement in the Therapeutic Efficacy of In Vivo BNCT Mediated by GB-10 with Electroporation in a Model of Oral Cancer

**DOI:** 10.3390/cells12091241

**Published:** 2023-04-25

**Authors:** Nahuel Olaiz, Andrea Monti Hughes, Emiliano C. C. Pozzi, Silvia Thorp, Paula Curotto, Verónica A. Trivillin, Paula S. Ramos, Mónica A. Palmieri, Guillermo Marshall, Amanda E. Schwint, Marcela A. Garabalino

**Affiliations:** 1Departamento de Física, Facultad de Ciencias Exactas y Naturales, Universidad de Buenos Aires (UBA), Pabellón I, Ciudad Universitaria, Buenos Aires C1428EHA, Argentina; 2Consejo Nacional de Investigaciones Científicas y Técnicas (CONICET), Godoy Cruz 2270, Buenos Aires C1425FQD, Argentina; 3Departamento Radiobiología, Comisión Nacional de Energía Atómica (CNEA), Centro Atómico Constituyentes (CAC), Av. General Paz 1499, San Martín, Buenos Aires B1650KNA, Argentina; 4Departamento de Reactores de Investigación y Producción, Comisión Nacional de Energía Atómica (CNEA), Centro Atómico Ezeiza (CAE), Camino Real Presbítero González y Aragón 15, Buenos Aires B1802AYA, Argentina; 5Sub-Gerencia Instrumentación y Control, Comisión Nacional de Energía Atómica (CNEA), Centro Atómico Ezeiza (CAE), Camino Real Presbítero González y Aragón 15, Buenos Aires B1802AYA, Argentina; 6Departamento de Biodiversidad y Biología Experimental, Facultad de Ciencias Exactas y Naturales, Universidad de Buenos Aires (UBA), Pabellón II, Ciudad Universitaria, Buenos Aires C1428EHA, Argentina

**Keywords:** oral cancer, hamster cheek pouch, BNCT, GB-10, electroporation, in vivo irradiation studies

## Abstract

Boron neutron capture therapy (BNCT) combines preferential tumor uptake of ^10^B compounds and neutron irradiation. Electroporation induces an increase in the permeability of the cell membrane. We previously demonstrated the optimization of boron biodistribution and microdistribution employing electroporation (EP) and decahydrodecaborate (GB-10) as the boron carrier in a hamster cheek pouch oral cancer model. The aim of the present study was to evaluate if EP could improve tumor control without enhancing the radiotoxicity of BNCT in vivo mediated by GB-10 with EP 10 min after GB-10 administration. Following cancerization, tumor-bearing hamster cheek pouches were treated with GB-10/BNCT or GB-10/BNCT + EP. Irradiations were carried out at the RA-3 Reactor. The tumor response and degree of mucositis in precancerous tissue surrounding tumors were evaluated for one month post-BNCT. The overall tumor response (partial remission (PR) + complete remission (CR)) increased significantly for protocol GB-10/BNCT + EP (92%) vs. GB-10/BNCT (48%). A statistically significant increase in the CR was observed for protocol GB-10/BNCT + EP (46%) vs. GB-10/BNCT (6%). For both protocols, the radiotoxicity (mucositis) was reversible and slight/moderate. Based on these results, we concluded that electroporation improved the therapeutic efficacy of GB-10/BNCT in vivo in the hamster cheek pouch oral cancer model without increasing the radiotoxicity.

## 1. Introduction

One of the challenges of oncological treatments is to damage tumor cells selectively, protecting normal tissue. Although advances have been made in therapeutic efficacy employing many various cancer therapies, new approaches are necessary to make tumors accessible and responsive. Boron neutron capture therapy (BNCT) coupled to electroporation (EP) would afford that possibility.

BNCT applies nuclear technology to biomedicine. It is a binary treatment that combines the administration of boron-10 carriers that are taken up preferentially by tumor cells and the subsequent irradiation with a thermal/epithermal neutron beam. The nuclear reaction between a ^10^B atom (a stable isotope) and a thermal neutron generates high linear energy transfer (LET) particles (alpha particles and ^7^Li recoil nuclei) and low LET gamma radiation [1,2,3]. High LET particles travel a short distance in tissue, approximately the diameter of a single tumor cell (10 μm); thus, they only destroy ^10^B-containing tumor cells [4], preserving the surrounding normal tissue. In this way, BNCT integrates the concepts of focusing in chemotherapy and of geometrical targeting in conventional radiotherapy [5].

BNCT clinical trials on recurrent glioblastoma, primary and recurrent head and neck cancer, sarcoma, meningioma, melanoma, hepatocellular carcinoma, and malignant mesothelioma were and/or are currently performed around the world [6,7]. Particularly, for the treatment of recurrent head and neck tumors, the BNCT overall tumor response rate was up to 90% [2].

Oral cancer is one of the most common types of cancer in the head and neck region [8]. The most well-known risk factors include tobacco consumption, frequent use of betel nut, alcohol, radiation, poor oral hygiene, genetic factors, and viruses such as HPV (human papilloma virus), among others [9,10]. As many patients seek medical attention with late-stage disease, they require invasive and expensive treatments impacting patient quality of life and increasing mortality [8,9,11]. Besides, antineoplastic therapies, including BNCT, frequently induce a common adverse effect called mucositis, an inflammatory reaction of the oral mucosa [12,13,14]. In this sense, there is a need for more effective and less toxic therapies to improve tumor control, patient survival, and quality of life [15].

Novel strategies that minimize mucositis and enhance BNCT therapeutic efficacy would provide encouraging improvements for patients. In this sense, studies in animal models will contribute to the advancement of BNCT [16,17]. The chemically induced hamster cheek pouch oral cancer model (Syrian hamster, *Mesocricetus auratus*) is widely used in the research on oral cancer and mucositis [18,19]. Our group proposed BNCT for the treatment of oral cancer in this experimental animal model [20,21], preceding the first BNCT clinical trials in head and neck cancer patients [22]. We evaluated, in this animal model, the therapeutic effect of BNCT mediated by different boron-10 carriers employing different administration protocols and studied strategies to increase BNCTs therapeutic effect and reduce adverse reactions, e.g., [23,24,25,26,27]. Employing the knowledge obtained in these preclinical radiobiological studies, we also demonstrated the therapeutic potential of BNCT to treat spontaneous head and neck tumors in veterinary patients without other therapeutic options [28,29,30].

The success of BNCT principally depends on a high absolute boron concentration values in tumors, high tumor/normal tissue (T/N) and tumor/blood (T/B) boron concentration ratios, and an effective boron microdistribution [31]. Our group has extensively studied sodium decahydrodecaborate (GB-10) as a potential boron carrier for BNCT. GB-10 was once approved for use in humans by the Food and Drug Administration of the USA [32]. In our studies, we demonstrated that although GB-10 is not incorporated selectively in tumors, BNCT mediated by GB-10 (GB-10/BNCT) induced a selective tumor effect by damaging the radiosensitive aberrant tumor vasculature [33].

Strategies currently approved and used in humans that could improve boron accumulation and microdistribution in tumor tissue would increase BNCTs therapeutic effect. In this sense, we proposed that electroporation (EP) would increase GB-10 uptake and favor microdistribution in tumor in the hamster cheek pouch oral cancer model [34]. This technique permeabilizes the cell membrane by applying an electric field, allowing the passage of molecules into the cytosol. It could be used as sole treatment (in the case of irreversible EP), inducing cell death, or as a strategy to deliver drugs (reversible EP) [35]. It has been used in clinical trials conducted in head and neck cancer patients [36] and in clinical veterinary patients with oral and maxillofacial tumors [35], and has been evaluated in in vitro and in vivo biodistribution studies with different boron carriers, e.g., [37,38,39,40,41]. In [34], we showed that GB-10 + early EP (10 min post administration of GB-10) induced a significant increase in the absolute and relative tumor boron concentration and optimized the boron microdistribution in the tumor, versus GB-10 without EP in the hamster cheek pouch oral cancer model.

Within this context, the aim of the present study was to evaluate the effect of EP coupled to BNCT with in vivo follow-up. During the 28 days after treatment, we assessed the therapeutic efficacy and associated radiotoxicity of GB-10/BNCT combined with early EP in the hamster cheek pouch oral cancer model.

## 2. Materials and Methods

For all protocols, we employed young Syrian hamsters (six to eight weeks old) exposed to the classical cancerization protocol: topical application of dimethyl-1,2-benzanthracene (DMBA, Sigma-Aldrich-Merck, Darmstadt, Germany) in mineral oil (0.5%) in the right cheek pouch twice a week for 12 weeks. The animals were maintained at a temperature below 24 °C with a 12/12 h light/dark cycle, with ad libitum tap water and a standard diet (Cooperación, Argentina). Cage changing was performed three times per week. The study was conducted according to the guidelines of the Declaration of Helsinki, National Institute of Health in the USA, regarding the care and use of animals for experimental procedures and approved by the Institutional Ethics Committee of the National Atomic Energy Commission of Argentina (CICUAL-CNEA, protocol codes 19/2018-20/2018, 7 November 2019).

The cancerized pouch was examined after cancerization and before and after irradiation to evaluate tumor development and mucositis (under light anesthesia, i.e., 140 mg/kg body weight ketamine and 21 mg/kg body weight xylazine injected intraperitoneally (i.p.)). We considered exophytic tumors as those that reached a volume of ≥1 mm^3^ and were ≥0.7 mm in height [42]. The tumor volume was calculated using an external caliper, considering the three largest orthogonal diameters (d) and calculating the tumor volume as d1 × d2 × d3, e.g., [43]. Two tumor volume ranges were defined arbitrarily, small (1 mm^3^ ≤ volume < 10 mm^3^) and medium/large (volume ≥ 10 mm^3^), to be able to detect potential differences in tumor response associated with tumor volume at the time of treatment.

### 2.1. GB-10 Solution

A stock solution of GB-10 (Na_2_^10^B_10_H_10_) (Neutron Therapies LLC, San Diego, CA, USA, Lot # NT000001, 2000), isotopically enriched to 99% in ^10^B (100 mg^10^B/mL, 1 M GB-10), was diluted 1:10 in distilled water to a final concentration of 10 mg^10^B/mL (equivalent to a 0.1 M solution of GB-10). The solution was sterilized with a 22 μm syringe filter and stored at 4 °C until use. GB-10 solution was administered intravenously (i.v.) in the surgically exposed jugular vein (under anesthesia) at a dose of 50 mg ^10^B/kg body weight (b.w.).

### 2.2. Electroporation

In our previous studies [34], we demonstrated in the hamster oral cancer model that electroporation 10 min after injection of GB-10 (early EP) was the best protocol in terms of tumor absolute boron uptake, the T/N and T/B ratios, and boron microdistribution in the tumor. In this study, we established the optimum conditions to electroporate tumors individually, employing two different plates depending on the tumor volume (Figure 1). For small tumors (<10 mm^3^), a 3 × 3 mm plate with a voltage between the two parallel metal plates of 200 V for a distance of 2 mm and 300 V for a distance of 3 mm was employed. For medium/large tumors (≥10 mm^3^), the plate was 10 × 10 mm in dimension, employing 400 V for a distance of 4 mm and 500 V for a distance of 5 mm. The animals were anesthetized (ketamine 140 mg/kg bw/xylazine 21 mg/kg bw, i.p.) to electroporate tumors individually. We used an electroporator (BTX ECM 830 Harvard Apparatus), employing the standard sequence of pulses for electrochemotherapy (1000 V/cm, 8 pulses of 100 μs) [44]. The electric current was measured with a 1-Ohm probe and a digital oscilloscope (InfniiVision DSO-X 2012A-SGM, Agilent Technologies, Santa Clara, CA, USA).

### 2.3. GB-10/BNCT Combined with Early Electroporation: In Vivo BNCT Studies

We evaluated five groups of cancerized tumor bearing hamsters:

GB-10/BNCT (50 mg ^10^B/kg bw) (t = 0 min) with early EP (t = 10 min after GB-10 injection) and irradiation (t = 3 h after GB-10 injection, 2:50 hs after EP) (n = 9 hamsters, 46 tumors);

GB-10/BNCT: GB-10 (50 mg ^10^B/kg bw) (t = 0 min) and irradiation (t = 3 h) (n = 5 hamsters, 33 tumors);

Only EP: EP (t = 0 min) (n = 5 hamsters, 38 tumors);

Beam only: neutron irradiation without GB-10 administration (n = 6 hamsters, 50 tumors);

Early EP + beam only: EP (t = 0 min) and irradiation (t = 2:50 hs after EP) (n = 5 hamsters, 32 tumors);

Control group: cancerized, not treated (data extracted from previous studies performed by our group [45] (n = 6 hamsters, 34 tumors).

The animals were irradiated at the RA-3 Nuclear Reactor thermal facility (Ezeiza, Buenos Aires, Argentina), employing a ^6^Li carbonate shield to protect the body of the animals from thermal neutrons, while everting the hamster cheek pouch bearing tumors onto a protruding shelf for exposure. All irradiations were performed at a thermal fluence of approximately 1.9 × 10^12^ n/cm^2^. We irradiated the animals 3 h after GB-10 injection on previous biodistribution studies performed by our group [23,34].

The thermal neutron fluence was matched for all the protocols. The absorbed doses for each tissue and protocol were calculated based on the absolute boron concentration values derived from previous biodistribution studies [34]. The absolute boron concentration values in tumor and precancerous tissues (the dose-limiting tissue) for the early EP + GB-10 group were 20.2 ± 9.6 ppm and 8.9 ± 1.1 ppm, respectively (tumor/precancerous tissue ratio = 2.3 ± 1.2). For the GB-10 group, the tumor and precancerous tissue absolute boron concentration values were 9.5 ± 2.4 ppm and 12.4 ± 1.4 ppm, respectively (tumor/precancerous tissue ratio = 0.8 ± 0.2) [34]. Based on these data for the GB-10 + early EP protocol, we estimated a total absorbed dose to precancerous tissue of approximately 2.1 ± 0.3 Gy and to tumors of 3.7 ± 1.5 Gy. For GB-10/BNCT, the estimated absorbed doses were 2.6 ± 0.3 Gy for precancerous tissue and 2.2 ± 0.6 Gy for tumors (Table 1). The irradiation time used for the beam only protocols (beam only and early EP + beam only) matched the irradiation time used in their corresponding BNCT groups (GB-10/BNCT and early EP + GB-10/BNCT). These irradiation times ranged between 238 and 276 s.

For 28 days from T0 (the day we applied each protocol), the clinical signs and body weight of the animals and the tumor response and mucositis in precancerous tissue were assessed. We considered (1) complete response (CR) as when tumors disappear on visual inspection; (2) partial response (PR) as when they undergo a reduction in their pre-treatment volume; (3) no response (NR) as when tumors grow or remain the same as at T0; and (4) overall response (OR) as partial response (PR) + complete response (CR).

Mucositis in precancerous tissue was evaluated semi-quantitatively employing an oral mucositis scale adapted from studies in hamsters and humans [45,46,47]: Grade 0: healthy appearance, no erosion or vasodilation; Grade 1 (slight): erythema and/or edema and/or vasodilation, no evidence of mucosal erosion; Grade 2 (slight): severe erythema and/or edema, vasodilation and/or superficial erosion; Grade 3 (moderate): severe erythema and/or edema, vasodilation, and the formation of ulcers <2 mm in diameter; Grade 4 (severe): severe erythema and/or edema, vasodilation, and the formation of ulcers ≥2 mm and <4 mm in diameter, and/or necrosis areas <4 mm in diameter; Grade 5 (severe): the formation of ulcers ≥4 mm in diameter or multiple ulcers ≥2 mm in diameter, and/or necrosis areas ≥4 mm in diameter. Grading was based on the most severe macroscopic feature.

The tumor response and the percentage of animals with severe mucositis were analyzed with a contingency table and Fisher’s exact tests. Statistical significance was set at *p* ≤ 0.05.

## 3. Results

The clinical status of the animals in terms of visual inspection and body weight showed no significant adverse effects after each protocol.

Figure 2 shows the therapeutic effect on tumors of each studied protocol at 28 days after T0. GB-10/BNCT + early EP significantly increased the percentage of overall tumor response vs. GB-10/BNCT from 48% to 92% (*p* < 0.0001) and complete responses from 6% to 46% (*p* = 0.0001). The percentage of non-responding tumors was reduced significantly (*p* < 0.0001) in the GB-10/BNCT + early EP group vs. GB-10/BNCT (8% vs. 52%, respectively). The tumor response for GB-10/BNCT + early EP (OR: 92%, CR: 46%) was significantly higher than for the other groups, i.e., EP only (OR: 33%, *p* < 0.0001, CR: 8%, *p* = 0.0002), beam only (OR: 18%, *p* < 0.0001, CR: 0%, *p*< 0.0001), beam only + early EP (OR: 38%, *p* < 0.0001, CR: 10%, *p* = 0.0009), and the control group (cancerized, not treated animals, OR: 21%, *p* = 0.0001, CR: 9%, *p* = 0.0004). EP alone induced some degree (albeit not statistically significant) of tumor response vs. the control group.

We also evaluated a possible correlation between tumor volume and the therapeutic effect induced by each protocol (Table 2). GB-10/BNCT + early EP significantly increased the percentages of OR vs. GB-10/BNCT for both small tumors (OR: 88% vs. 40%, *p* = 0.0030) and medium and large tumors (95% vs. 56%, *p* = 0.0067). It is interesting that EP also significantly increased the tumor complete responses induced by GB-10/BNCT in small tumors (from 7% for GB-10/BNCT to 65% for GB-10/BNCT + early EP, *p* = 0.0003). In the case of medium and large tumors, an increase in CR was observed for the GB-10/BNCT + early EP protocol (20%) vs. GB-10/BNCT (6%). However, this difference did not reach statistical significance. EP also increased OR in tumors treated with beam only (*p* = 0.0195), with some tumor complete responses in the small tumors group.

Figure 3 represents one of our best examples of complete and partial tumor responses and mucositis resolution after GB-10/BNCT + early EP during 28 days follow-up.

Related to mucositis in the dose-limiting tissue, the precancerous tissue around tumors, we observed that none of the BNCT protocols induced severe mucositis. For all cases, mucositis resolved to grade 2 or less at 28 days after T0 (Table 3 and Figure 3).

## 4. Discussion

In this study, we performed an in vivo follow-up of 28 days after EP combined with BNCT, and demonstrated that EP increases the therapeutic effect of BNCT mediated by GB-10 in the hamster cheek pouch oral cancer model. The animals exhibited only slight/moderate and reversible mucositis after treatment and were in good health after 28 days of follow up.

Previous studies by other groups suggested that EP might be therapeutically useful in BNCT. Several groups evaluated the effect of EP on boron accumulation in the tumor, employing different boron compounds, but did not study the actual effect of BNCT combined with EP in tumor bearing animals [37,38,39,40,41]. Ono et al. (2000) [41] studied the effect of EP combined with BNCT mediated by sodium borocaptate (BSH) on excised tumors derived from implanted SCCVII tumor cells (mouse squamous cell carcinoma) in mice. BSH is a diffusive agent, used as a boron carrier for BNCT for malignant glioma. Its selectivity depends on the disrupted blood–brain barrier in these tumors, and it exhibits low accumulation in tumor cells. In work by Ono et al. (2000) [41], tumor-bearing animals were injected with BSH. Tumors were electroporated 15 min after BSH administration, and 3 h later the tumors were surgically removed. These tumors were irradiated inside closed Teflon tubes and 5 min after irradiation colony formation assays were performed. EP improved the cell killing effect by BSH/BNCT in tumor cells.

GB-10 is a diffusive boron carrier, similar to BSH. In previous studies with GB-10, we observed that boron did not show a preferential accumulation in tumors. However, GB-10/BNCT did have a selective effect on tumors by selectively damaging the more radiosensitive aberrant tumor blood vessels [33]. Although this experiment was quite encouraging, as GB-10/BNCT induced only slight mucositis in precancerous tissue (the dose-limiting tissue) compared to BPA/BNCT, which induces moderate to severe mucositis, the therapeutic effect on tumors had to be improved [27,33,43]. In the present study, we also showed that GB-10/BNCT without EP (at the neutron fluence employed) induced an OR of 48% of the tumors, with 6% of the tumors exhibiting a CR. In that sense, a strategy that could improve boron accumulation and microdistribution in tumor tissue would increase GB-10/BNCTs therapeutic effect.

The use of strategies and boron compounds approved for use in humans will help to bridge the gap between research and clinical studies [48,49]. In this sense, GB-10 was once approved for its use in humans by the FDA and EP is currently used in humans and veterinary patients as a sole treatment or as a compound delivery strategy [32,35]. In wour previous work [34], we employed GB-10 and compared two different EP protocols: late (2:50 h after GB-10 injection, i.e., 10 min before irradiation) and early EP (10 min after GB-10 injection, i.e., 2:50 h before irradiation). We demonstrated that early EP significantly increased the absolute boron concentration and boron targeting homogeneity in the tumor at 3 h post GB-10. In previous studies, we also showed that the boron concentration in blood 10 min after GB-10 i.v. injection was four times higher than at 3 h after injection [23]. This finding would support the use of EP 10 min after administration of GB-10 to maximize the uptake from blood to tumor. It is known that EP induces an anti-vascular effect called “vascular lock” that lasts at most for 10 min, followed by an increase in vascular permeability [50]. The vascular lock effect reduces the blood flow in tumors without affecting the vasculature in adjacent tissue [51,52]. These effects would contribute to improving the retention of chemotherapeutics in tumors [52]. In this study, the significantly higher absolute boron concentration values in tumors treated with GB-10 + early EP could be due to a higher availability of GB-10 in blood at the time of EP. When EP was performed, the induced vascular lock would lead to an increase in boron concentration in tumor stroma. As EP increases tumor cell permeability, it enhances GB-10 uptake by tumor cells, thus increasing boron targeting homogeneity in the tumor (as shown in the qualitative autoradiography studies in [34]). Furthermore, EP is known to exert an anti-angiogenic effect, selectively destroying small tumor blood vessels without affecting the larger blood vessels surrounding the tumor [51]. These effects would explain the significant increase in the therapeutic effect in terms of the overall tumor responses and complete responses in the GB-10/BNCT + early EP group compared to GB-10/BNCT group.

We observed that GB-10/BNCT + early EP increased the percentage of CR in small tumors. The electric pulses can reach all tumor cell populations more easily in small tumors, while in medium and large tumors, tumor cells at the center of the tumor could be receiving less electricity [53,54]. Previous studies have also shown that the antitumor efficacy of EP used for the delivery of chemotherapeutic agents (electrochemotherapy—ECT) was inversely proportional to the tumor size. These authors compared ECT and surgery in dogs with mast cell tumors (MCT). They observed that ECT induced a 70% complete response rate, while surgery induced a 50% complete response rate. They proposed ECT as an alternative treatment to surgery, especially in cases of smaller MCTs and those that are unresectable due to their location [55]. For larger tumors, they proposed a combination of surgical removal and intraoperative ECT [35,56].

Since EP is applied only to tumors, cell permeability does not vary significantly in non-tumor tissue. We observed that the absolute boron concentration values in blood and precancerous tissue were similar for the GB-10 and GB-10 + EP groups [34]. This would explain why mucositis in the GB-10/BNCT + early EP was slight/moderate and similar to the GB-10/BNCT group, even though the tumor response increased significantly in the GB-10/BNCT + early EP group. This result is of outmost importance, as we were able to improve the therapeutic effect of GB-10/BNCT and preserve the dose-limiting tissue. Future studies will be aimed at increasing the absorbed dose to precancerous tissue to obtain higher absorbed doses in the tumor and improve tumor control. Particularly, this would help to improve complete responses in the medium + large tumors group. The fact that mucositis in precancerous tissue was slight/moderate and reversible would also allow for re-treatment with GB-10/BNCT + EP as a strategy to improve the tumor response.

We also observed some degree of tumor response with EP alone. In addition, EP increased the beam only (background dose) therapeutic effect on tumors. It is known that EP can induce, depending on the time of exposure to electrical pulses and strength of the electric field, a reversible increase in permeability (generally used for the delivery of therapeutic agents) or irreversible permeability that induces immunogenic response stimulation [57,58].

EP and BNCT are both local cancer therapies. As EP and BNCT were shown to trigger immunogenic cell death, e.g., [59,60,61,62], an interesting approach would be to combine EP + BNCT with immunotherapy to enhance this effect and offer a systemic therapy to patients suffering from diffuse tumors, hematological malignancies, and/or metastases that do not respond to standard therapies.

The EP technique significantly optimized the therapeutic effect of BNCT mediated by GB-10, without enhancing the radiotoxic effects in the dose-limiting tissue. These results suggest it might be promising to consider BNCT combined with EP in a clinical scenario for head and neck cancer patients.

## Figures and Tables

**Figure 1 cells-12-01241-f001:**
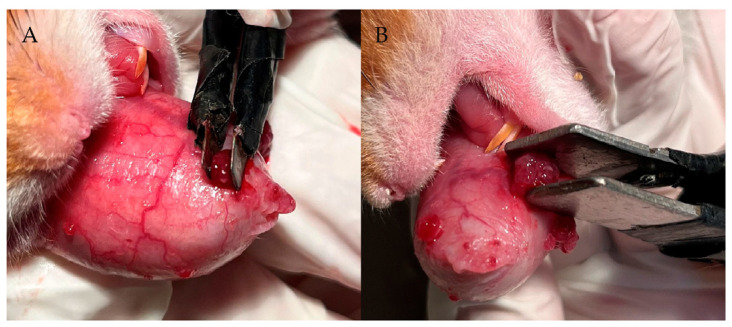
Electroporation plates size and conditions depending on tumor volume. (**A**) Small tumors (<10 mm^3^): 3 × 3 mm plates, 200 V for a distance of 2 mm, and 300 V for a distance of 3 mm; (**B**) Medium/large tumors (≥10 mm^3^): 10 × 10 mm plates, 400 V for a distance of 4 mm and 500 V for a distance of 5 mm.

**Figure 2 cells-12-01241-f002:**
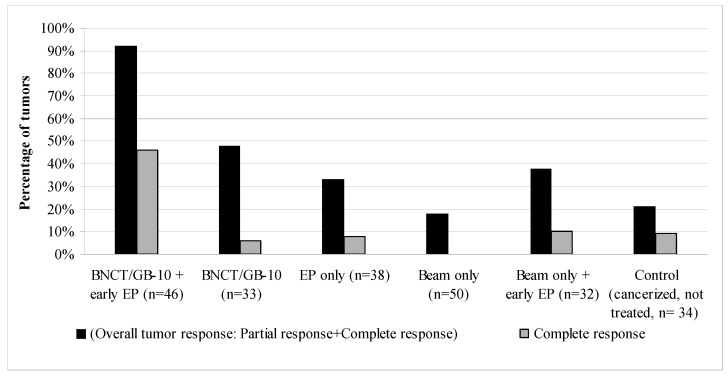
Percentage of tumors with an overall tumor response (OR = PR + CR) and a complete response (CR) at 28 days after T0 for each of the studied protocols. n: number of tumors.

**Figure 3 cells-12-01241-f003:**
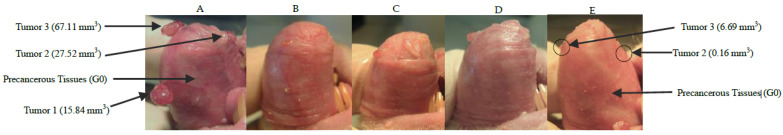
Macroscopic follow up of the hamster cheek pouch bearing tumors treated with GB-10/BNCT + early EP at T0 (**A**), 7 days (**B**), 14 days (**C**), 21 days (**D**), and 28 days (**E**) after treatment. The hamster cheek pouch exhibited slight mucositis that completely resolved by the end of the follow-up. (**E**) Tumor 1 exhibited a complete response (CR); Tumors 2 and 3 exhibited volume reductions of almost 90% after 28 days after GB-10/BNCT + early EP. G0 = Grade 0 mucositis.

**Table 1 cells-12-01241-t001:** Absorbed doses (Gy) for each of the radiation components in precancerous tissue and tumors corresponding to GB-10/BNCT and GB-10/BNCT + early EP protocols. Boron concentration (ppm) values for tumor and precancerous tissue used for dose calculations. The mean thermal neutron fluence at the irradiation position was 1.9 × 10^12^ n/cm^2^ and the irradiation time ranged from 238 to 276 s. Values are presented as means ± SD.

Protocol	Tissue	ppm [^10^B]	Induced Protons (^14^N)	Total γ Ray Dose	Boron Absorbed Dose	Total Absorbed Dose
GB-10/BNCT	Precancerous tissue	12.4 ± 1.4	0.40 ± 0.09	0.41 ± 0.05	1.76 ± 0.4	2.6 ± 0.3
	Tumor	9.5 ± 2.4	0.40 ± 0.09	0.41 ± 0.05	1.35 ± 0.4	2.2 ± 0.6
GB-10/BNCT + early EP	Precancerous tissue	8.9 ± 1.1	0.40 ± 0.09	0.41 ± 0.05	1.26 ± 0.3	2.1 ± 0.3
	Tumor	20.2 ± 9.6	0.40 ± 0.09	0.41 ± 0.05	2.86 ± 1.5	3.7 ± 1.5

**Table 2 cells-12-01241-t002:** Percentage (%) of tumors with partial response (PR), complete response (CR), no response (NR), and overall tumor control (OR = PR + CR) at 28 days after T0. Tumors were classified depending on their initial volume: S—small tumors, volume <10 mm^3^; M + L—medium and large tumors, volume ≥10 mm^3^. (N = number of tumors.).

	GB-10/BNCT + Early EP		GB-10/BNCT		EP Only		Beam Only		Beam Only + Early EP		Control (Cancerized, Not Treated)	
	S	M + L	S	M + L	S	M + L	S	M + L	S	M + L	S	M + L
PR	23%	75%	33%	50%	18%	33%	18%	17%	14%	50%	11%	6%
CR	65%	20%	7%	6%	18%	0%	0%	0%	18%	0%	6%	6%
NR	12%	5%	60%	44%	64%	67%	82%	83%	68%	50%	83%	88%
OR: PR + CR	88%	95%	40%	56%	36%	33%	18%	17%	32%	50%	17%	12%
N	26	20	15	18	17	21	38	12	22	10	18	16

**Table 3 cells-12-01241-t003:** Percentage (%) of animals with mucositis in precancerous tissue for each of the studied protocols. We analyzed the maximum mucositis (reached at any point during follow up) and mucositis resolution at 28 days (end of the follow up). (n: number of hamsters, G0–G2: slight mucositis, G3: moderate mucositis, G4–G5: severe mucositis).

Mucositis in Precancerous Tissue		BNCT/GB-10 + Early EP	BNCT/GB-10	EP Only	Beam Only	Early EP + Beam Only	Control (Cancerized, Not Treated)
Maximum Mucositis	G0–G2	89%	100%	100%	100%	80%	100%
	G3	11%	0%	0%	0%	0%	0%
	G4–G5	0%	0%	0%	0%	20%	0%
Mucositis resolution after 28 days	G0–G2	100%	100%	100%	100%	100%	100%
	G3	0%	0%	0%	0%	0%	0%
	G4–G5	0%	0%	0%	0%	0%	0%
	N	9	5	5	6	5	6

## Data Availability

Not applicable.

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
