# Peer review of "Enhancement in the Therapeutic Efficacy of In Vivo BNCT Mediated by GB-10 with Electroporation in a Model of Oral Cancer"

_cells, 2023, doi:10.3390/cells12091241_

Round 1

Reviewer 1 Report

The radical vision is clear in this paper which is to expand translational research on new therapies for tumours, and to hopefully launch new clinical trials. Maximizing therapeutic effectiveness necessitates using many treatment modalities at once, which here is shown.

minor concerns

1) On page 4 

Line 164 how long is the time for Beam only

Line 166 why is "Early EP + beam only t=2:50 hs after EP".

Why choosing 3 hrs? does it depend on tumor size?

2) On Page 5 Line 207-209 "GB-10/BNCT + early EP increased significantly the percentage of overall tumor 207 responses vs. GB-10/BNCT from 48% to 92% (p<0.0001) and complete responses from 6% 208 to 46% (p=0.0001). " Does it happen with reduced time of exposure? What happens if the authors  changed standard sequence of pulses for electrochemotherapy? 

As a visual point of view, please fix figures which are not aligned.

Author Response

April 16th, 2023

Dear Reviewer,           

I am submitting to your consideration for publication in CELLS – Special Issue “BNCT Drug Development and Preclinical Testing” the revised version of our article entitled “ENHANCEMENT IN THE THERAPEUTIC EFFICACY OF IN VIVO BNCT MEDIATED BY GB-10 WITH ELECTROPORATION IN A MODEL OF ORAL CANCER”. We have addressed all the suggestions made by you. We will also upload the manuscript in track changes mode.

Hoping you will find our manuscript suitable for publication,

Yours sincerely,

Marcela A. Garabalino, PhD

Dept. Radiobiology, National Atomic Energy Commission (CNEA)

Avenida General Paz 1499, B1650KNA San Martin

Provincia Buenos Aires, Argentina

Tel.: +5491167727148; e-mail: garabalino@cnea.gov.ar / marcegarabalino@gmail.com

Reviewer 1: Thank you very much for your revision. We appreciate all your comments and suggestions. Please find below all the answers.

1) On page 4, Line 164: how long is the time for Beam only.

The irradiation time used for the Beam only protocols (Beam only, early EP+Beam only) match the irradiation time used in for their corresponding BNCT groups (GB-10/BNCT, early EP+GB-10/BNCT). These irradiation times ranged between 238 to 276 seconds.

2) Line 166 why is "Early EP + beam only t=2:50 hs after EP".

"Early EP + beam only" is the control group associated to the protocol “Early EP+GB-10/BNCT”. Our aim was to evaluate the effect of EP and the background dose (no boron dose component) on tumor control and toxicity in these conditions. In this sense, we reproduced exactly the protocol but without injecting GB-10. To clarify: The early EP+GB-10/BNCT group consisted of injecting GB-10, after 10 minutes we electroporated the tumors, and then, after 2:50h, we irradiated the cancerized pouches. For the early EP+Beam only group, NO GB-10 injection was performed, so, we electroporated the tumors and, after 2:50 hours, we irradiated the cancerized pouches bearing tumors.

3) Why choosing 3 hrs? does it depend on tumor size?

The selected irradiation time, 3 hours after the GB-10 administration, is based on previous boron biodistribution studies with GB-10 in the hamster cheek pouch oral cancer model performed by our work group, cited in this work as Reference 23. The time selected does not depend on tumor size.

4) On Page 5 Line 207-209 "GB-10/BNCT + early EP increased significantly the percentage of overall tumor responses vs. GB-10/BNCT from 48% to 92% (p<0.0001) and complete responses from 6% to 46% (p=0.0001)." Does it happen with reduced time of exposure? What happens if the authors changed standard sequence of pulses for electrochemotherapy?

In previous non-published pilot studies (2 hamsters, 11 tumors) we analyzed the effect of Early EP on GB-10 biodistribution employing 1000 V/cm, 4 pulses of 100 μs instead or 8 pulses (as reported in our manuscript). Boron concentration values for blood, tumor, precancerous and normal tissue (see concentration values below this paragraph) resulted similar to those values observed, in these tissues, employing GB-10 alone (reference 34). This result indicated that Early EP, in these conditions, did not enhance boron uptake by the tumor, and thus would not enhance BNCT tumor control. Besides, these ECT parameters employed in our manuscript (1000 V/cm, 8 pulses of 100 μs) are used in ECT clinical studies in humans: Michel Marty, Gregor Sersa, Jean Rémi Garbay, Julie Gehl, Christopher G. Collins, Marko Snoj, Valérie Billard, Poul F. Geertsen, John O. Larkin, Damijan Miklavcic, Ivan Pavlovic, Snezna M. Paulin-Kosir, Maja Cemazar, Nassim Morsli, Declan M. Soden, Zvonimir Rudolf, Caroline Robert, Gerald C. O’Sullivan, Lluis M. Mir. Electrochemotherapy – An easy, highly effective and safe treatment of cutaneous and subcutaneous metastases: Results of ESOPE (European Standard Operating Procedures of Electrochemotherapy) study, European Journal of Cancer Supplements, Volume 4, Issue 11, 2006, Pages 3-13. doi.org/10.1016/j.ejcsup.2006.08.002). This reference was included in our mansucript (Reference number 44).

Boron concentration values for the protocol “EarlyEP+GB-10, 1000 V/cm, 4 pulses of 100 μs” (data not published): Blood (15.0 ± 0.8, n=2), precancerous tissue (10.8± 2.3, n=2), normal tissue (9.9±3.7) and Tumor (8.1±2.4, n=11).

5) As a visual point of view, please fix figures which are not aligned.

The figures were aligned. Thank you.

Reviewer 2 Report

This paper reports the evaluation of the therapeutic efficacy of in vivo boron neutron capture therapy (BNCT) coupled with early electroporation (EP) in the hamster cheek pouch oral cancer model.  The results suggest that BNCT coupled to electroporation improves the overall tumor response and complete tumor response compared to that of BNCT alone, while still remains slight/moderate radiotoxicity (mucositis).  Most of the results are straightforward and consistent, although the following revisions should be considered:

(1) Please replace “9 animals, 23 tumors” by “n=9 hamsters, 23 tumors” on Page 4 line 160 to be consistent with description in other treatment groups.

(2) The number of tumors in the Methods section do not match the values shown in the Results section.  On page 4 line 160, number of tumors is 23 for BNCT/GB-10 + early EP group.  However, the value is 46 in Figure 2 and Table 1.  The same problem applies for the Beam only group as well.  Please check the numbers.

(3) Please clarify how were obtained the total absorbed dose in GB-10/BNCT + early EP group for 2.1 Gy (precancerous tissue) and 3.7 Gy (tumor) since these values were estimated from the boron concentration.  The values for GB-10/BNCT group corresponds well with the ratio shown in the description (2.2/2.6 = 0.8), but not for the GB-10/BNCT + early EP group.  Please include the standard deviation for these two values as well.

(4) Please remake Figure 3.  The legends are not informative.  I do not think the arrows are pointing towards the right area.

(5) Please consider adding the data from control group (not treated) to Table 1 and Table 2 as comparison, since the control group data are shown in Figure 2.

(6) I don’t think it is appropriate to write “For the first time,” which is a self-congratulatory statement.  And it is certainly not the best way to start a Discussion section.  If something is a first, the scientific community will recognize it.  The authors don’t need to call this out; it sounds like braggadocio.

In all, this is an interesting paper that should be published with minor revisions, and should be of significant impact for this particular type of cancer, and perhaps others.

Author Response

April 16th, 2023

Dear Reviewer,           

I am submitting to your consideration for publication in CELLS – Special Issue “BNCT Drug Development and Preclinical Testing” the revised version of our article entitled “ENHANCEMENT IN THE THERAPEUTIC EFFICACY OF IN VIVO BNCT MEDIATED BY GB-10 WITH ELECTROPORATION IN A MODEL OF ORAL CANCER”. We have addressed all the suggestions made by you. We will also upload the manuscript in track changes mode.

Hoping you will find our manuscript suitable for publication,

Yours sincerely,

Marcela A. Garabalino, PhD

Dept. Radiobiology, National Atomic Energy Commission (CNEA)

Avenida General Paz 1499, B1650KNA San Martin

Provincia Buenos Aires, Argentina

Tel.: +5491167727148; e-mail: garabalino@cnea.gov.ar /marcegarabalino@gmail.com

Reviewer 2: Thank you very much for your revision. We appreciate all your comments and suggestions. Please find below all the answers to your revision.

(1)       Please replace “9 animals, 23 tumors” by “n=9 hamsters, 23 tumors” on Page 4 line 160 to be consistent with description in other treatment groups.

We changed the text as requested. Thank you.

(2)       The number of tumors in the Methods section do not match the values shown in the Results section.  On page 4 line 160, number of tumors is 23 for BNCT/GB-10 + early EP group.  However, the value is 46 in Figure 2 and Table 1.  The same problem applies for the Beam only group as well.  Please check the numbers.

Thank you for your comment. We checked and corrected the number of tumors as requested.

(3)       Please clarify how were obtained the total absorbed dose in GB-10/BNCT + early EP group for 2.1 Gy (precancerous tissue) and 3.7 Gy (tumor) since these values were estimated from the boron concentration.  The values for GB-10/BNCT group corresponds well with the ratio shown in the description (2.2/2.6 = 0.8), but not for the GB-10/BNCT + early EP group.  Please include the standard deviation for these two values as well.

Thank you for your suggestion. To clarify this point, we added a table (Table 1) with the irradiation conditions (i.e. absorbed dose for each component and total absorbed dose) for each tissue in each protocol.

(4)       Please remake Figure 3.  The legends are not informative.  I do not think the arrows are pointing towards the right area.

Thank you for your suggestion. We corrected the figure as requested.

(5)       Please consider adding the data from control group (not treated) to Table 1 and Table 2 as comparison, since the control group data are shown in Figure 2.

Data from the control group (not treated) were added in Tables 2 and 3. We renumbered the tables of our manuscript according to suggestion number 3 (Table 1. Irradiation conditions).

(6) I don’t think it is appropriate to write “For the first time,” which is a self-congratulatory statement.  And it is certainly not the best way to start a Discussion section.  If something is a first, the scientific community will recognize it.  The authors don’t need to call this out; it sounds like braggadocio.

In all, this is an interesting paper that should be published with minor revisions, and should be of significant impact for this particular type of cancer, and perhaps others.

Thank you for your comment. We adapted the phrase in the discussion section as requested.